# Octyl Gallate as an Intervention Catalyst to Augment Antifungal Efficacy of Caspofungin

**Jong H. Kim \*, Kathleen L. Chan and Luisa W. Cheng**

Foodborne Toxin Detection and Prevention Research Unit, Western Regional Research Center, USDA-ARS, 800 Buchanan St., Albany, CA 94710, USA; kathy.chan@ars.usda.gov (K.L.C.); luisa.cheng@ars.usda.gov (L.W.C.)
\* Correspondence: jongheon.kim@ars.usda.gov; Tel.: +1-510-559-5841

**Abstract:** Filamentous fungi such as *Aspergillus* spp. are opportunistic pathogens, which cause highly invasive infections, especially in immunocompromised individuals. Control of such fungal pathogens is increasingly problematic due to the small number of effective drugs available for treatment. Moreover, the increased incidence of fungal resistance to antifungal agents makes this problem a global human health issue. The cell wall integrity system of fungi is the target of antimycotic drugs echinocandins, such as caspofungin (CAS). However, echinocandins cannot completely inhibit the growth of filamentous fungal pathogens, which results in survival/escape of fungi during treatment. Chemosensitization was developed as an alternative intervention strategy, where co-application of CAS with the intervention catalyst octyl gallate (OG; chemosensitizer) greatly enhanced CAS efficacy, thus achieved ≥99.9% elimination of filamentous fungi in vitro. Based on hypersensitive responses of *Aspergillus* antioxidant mutants to OG, it is hypothesized that, besides destabilizing cell wall integrity, the redox-active characteristic of OG may further debilitate the fungal antioxidant system.

**Keywords:** antioxidant system; cell wall integrity; chemosensitization; end point; fungi; small molecule

---

## 1. Introduction

Fungal infectious diseases, such as candidiasis, cryptococcosis or invasive aspergillosis caused by *Candida*, *Cryptococcus* or *Aspergillus*, respectively, are serious human health issues, since effective drugs, especially those for eliminating resistant pathogens, are often very limited ([1] and the references therein). Therefore, there is continuous need to improve the efficacy of current antifungal drugs or to discover/develop new intervention strategies. The cell wall integrity system of fungal pathogens could serve as an effective target of antimycotic drugs [2]. Genome and functional studies revealed that many genes in the cell wall integrity system of fungi are well conserved [3,4]. Caspofungin (CAS; Figure 1), like other echinocandins including micafungin and anidulafungin, is an antifungal lipopeptide drug. CAS is in clinical use due to its good solubility, antifungal spectrum and pharmacokinetic properties. CAS inhibits the activity of β-1,3-D-glucan synthase in the fungal cell wall integrity system, thus disrupting the synthesis of the cell wall component β-1,3-D-glucan ([5] and the references therein). Echinocandins further lyse actively-growing hyphal tips during filamentous fungal growth [5]. However, despite their utility, echinocandins generally cannot achieve complete inhibition of the growth of filamentous fungi [6], which results in pathogen survival during treatment.

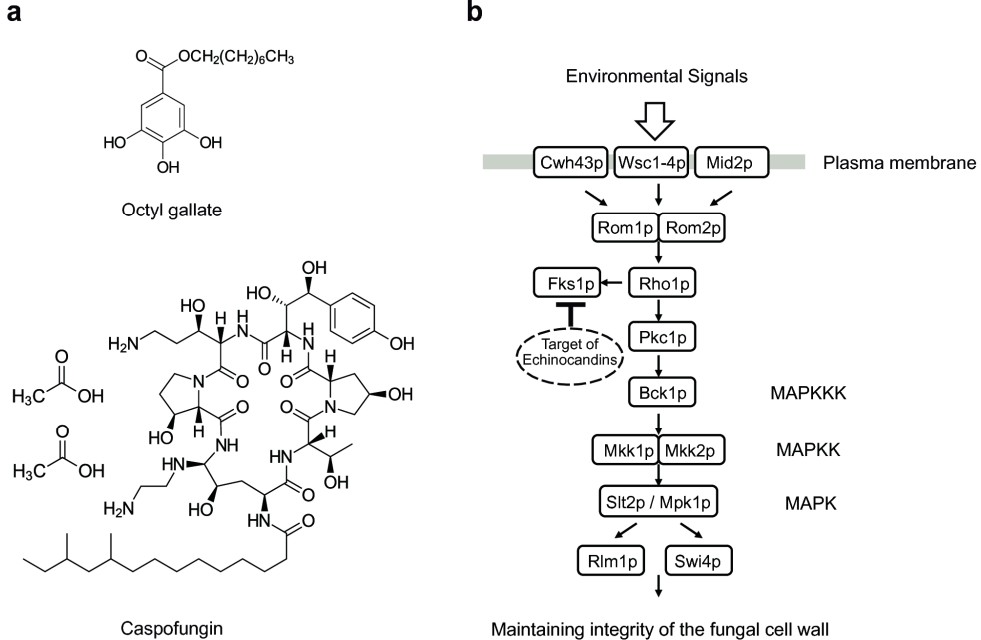

**Figure 1.** (**a**) Structures of octyl gallate (OG) and caspofungin (CAS); (**b**) signal transduction pathway of fungi for maintaining cell wall integrity, viz. sensing the status of the cell wall during growth and/or protecting the cell from environmental cues, such as external osmotic fluctuation (see [4,7] and the references therein). See Table S1 for the functions of proteins in the pathway, except: Rom1p, GDP/GTP exchange protein for Rho1p; Cwh43p, putative sensor/transporter protein involved in cell wall biogenesis; Rho1p, GTP-binding protein, which regulates protein kinase C (Pkc1p) and the cell wall synthesizing enzyme β-1,3-D-glucan synthase (Fks1p; target of echinocandins).

Antifungal chemosensitization is an intervention strategy, where co-application of a certain natural or synthetic compound, viz. chemosensitizer (intervention catalyst), with a commercial drug greatly enhances the efficacy of the co-applied drug [8]. The key advantage of chemosensitization is that, in contrast to combination therapy, which is a co-application of two or more commercial antimycotic drugs, a chemosensitizer itself does not have to possess a high degree of antifungal potency. However, chemosensitization not only enhances the antifungal efficacy of the co-applied drug, but also mitigates pathogen resistance to conventional drugs [8]. Therefore, chemosensitization-based intervention could complement current antifungal practices, such as combination therapy.

As a proof of concept, the effectiveness of antifungal chemosensitization (CAS + octyl gallate (OG; octyl 3,4,5-trihydroxybenzoic acid)) was investigated in the species of *Aspergillus* and *Penicillium* in this study (Table 1). Test strains belong to clinical and foodborne fungal pathogens, including environmental fungal contaminants. OG, an alkyl derivative of the natural product gallic acid (Figure 1), was investigated as a potent chemosensitizer (intervention catalyst) to enhance the efficacy of cell wall-disrupting drug CAS. We recently determined that OG functioned as a safer, more effective preservative for consumer products [9].

**Table 1.** Filamentous fungal strains used in this study.

| *Aspergillus* Strains | Strain Characteristics | Source |
|---|---|---|
| *A. fumigatus* AF293 | Human pathogen (aspergillosis), parental strain, reference clinical strain used for genome sequencing | [10] |
| *A. fumigatus* sakAΔ | Mitogen-activated protein kinase (MAPK) gene deletion mutant derived from AF293 | [10] |
| *A. fumigatus* mpkCΔ | MAPK gene deletion mutant derived from AF293 | [11] |
| *A. flavus* 3357 | Toxigenic (aflatoxin-producing), human pathogen (aspergillosis), reference strain for genome sequencing | NRRL [1] |
| *A. flavus* 4212 | Toxigenic (aflatoxin-producing), human pathogen (aspergillosis) | NRRL |
| *A. parasiticus* 2999 | Toxigenic (aflatoxin-producing) | NRRL |
| *A. parasiticus* 5862 | Toxigenic (aflatoxin-producing) | NRRL |
| *Penicillium* Strains | Strain Characteristics | Source |
| *P. expansum* W1 | Toxigenic (patulin-producing; parental strain) | [12] |
| *P. expansum* FR2 | Fludioxonil resistant mutant derived from *P. expansum* W1 | [12] |
| *P. expansum* W2 | Toxigenic (patulin-producing; parental strain) | [12] |
| *P. expansum* FR3 | Fludioxonil resistant mutant derived from *P. expansum* W2 | [12] |
| *P. glabrum* 766 | Environmental contaminant | NRRL |
| *P. chrysogenum* 824 | Fleming's penicillin-producing strain | NRRL |
| *P. griseofulvum* 2159 | Environmental contaminant | NRRL |
| *P. italicum* 983 | Environmental contaminant | NRRL |

[1] NRRL, National Center for Agricultural Utilization and Research, USDA-ARS, Peoria, IL, USA.

## 2. Materials and Methods

### 2.1. Chemicals

All chemicals including antifungal compounds (caspofungin (CAS), octyl gallate (OG)) and culture media were procured from Sigma Co. (St. Louis, MO, USA). CAS and OG were dissolved in dimethylsulfoxide (DMSO; absolute DMSO amount: <2% in media) before incorporation into culture media. Throughout this study, control plates (no treatment) contained DMSO at levels equivalent to that of cohorts receiving antifungal agents, within the same set of experiments.

### 2.2. Antifungal Bioassay: Saccharomyces cerevisiae

Susceptibility of the model yeast *S. cerevisiae* (See Table S1) was tested according to the protocol outlined by European Committee on Antimicrobial Susceptibility Testing (EUCAST) for yeasts [13]. Quantitative 96-well microtiter plate broth-dilution assays were performed in triplicate in liquid synthetic glucose (SG; yeast nitrogen base without amino acids 0.67%, glucose 2% with appropriate supplements: uracil 0.02 mg/mL, amino acids 0.03 mg/mL) medium, where the minimum inhibitory concentration (MIC; lowest concentration of compound showing no visible fungal growth in microtiter wells (200 µL per well)) was assessed after 24 h at 30 °C. Minimum fungicidal concentration (MFC; lowest concentration of compound showing ≥99.9% death of fungal cells inoculated, viz. achievement of ≥3 log fungal elimination) was determined after completion of MIC assays by spreading entire volumes of microtiter wells (200 µL) onto individual yeast peptone dextrose (YPD; Bacto yeast extract 1%, Bacto peptone 2%, glucose 2%) recovery plates. Colony-forming units were counted after additional incubation of plates for 48 h at 30 °C.

## 2.3. Antifungal Bioassay: Filamentous Fungi

To determine the level of compound interactions, namely the chemosensitizing activity of OG to CAS, in filamentous fungi (Table 1), triplicate checkerboard bioassays ($4 \times 10^3$–$5 \times 10^4$ CFU/mL) were performed in 96-well microtiter plates at 28 or 35 °C, depending on the types of strains, using a broth microdilution method in RPMI 1640 medium (Sigma Co., St. Louis, MO, USA) according to the protocol described by the Clinical and Laboratory Standards Institute (CLSI) M38-A [14]. MICs of antimycotic compounds, alone or in combination, were assessed after 48 h. MFCs of CAS and OG, alone or in combination, were determined following the completion of MIC analysis by spreading entire volumes of microtiter wells (200 μL) onto individual potato dextrose agar (PDA) recovery plates and culturing for an additional 48 h. Compound interactions, namely fractional inhibitory concentration indices (FICIs) and fractional fungicidal concentration indices (FFCI) for determining CAS + OG synergism for "growth inhibitory" and "fungal death", respectively, were calculated as follows: FICI or FFCI = (MIC or MFC of Compound A in combination with Compound B/MIC or MFC of Compound A, alone) + (MIC or MFC of Compound B in combination with Compound A/MIC or MFC of Compound B, alone). Levels and types of compound interactions between antimycotic agents were defined as: synergistic (FICI or FFCI ≤ 0.5) or indifferent (FICI or FFCI > 0.5–4) [15]. If preferred, Isenberg's [16] methodology could be substituted in parallel determinations of synergism, where compound interactions were: synergistic (FICI ≤ 0.5), additive (0.5 < FICI ≤ 1), neutral (1 < FICI ≤ 2) or antagonistic (FICI > 2).

## 2.4. Statistical Analysis

Statistical analysis (Student's *t*-test) was performed according to "Statistics to use" [17], where $p < 0.05$ was considered significant.

## 3. Results and Discussion

### 3.1. Octyl Gallate Perturbs the Fungal Cell Wall Integrity System: S. cerevisiae Bioassay

To determine whether OG could target the cell wall integrity system of fungi, OG susceptibility of eleven mutants of the model yeast *S. cerevisiae*, where genes in the cell wall integrity mitogen-activated protein kinase (MAPK) pathway were systematically deleted (Figure 1, Table S1), was initially examined. *S. cerevisiae* is a useful model system for identifying antifungal agents and their gene targets in that: (1) many genes in *S. cerevisiae* are orthologs of genes of fungal pathogens [18]; and (2) *S. cerevisiae* gene deletion mutant collections have proven to be very useful for genome-wide drug-induced haploinsufficiency screens to determine drug's mode of action [19–21]. OG is a generally recognized as safe (GRAS) reagent [22] and, thus, is currently used as an antioxidant added to food. OG is also known to inhibit the growth of bacterial pathogens, such as *Staphylococcus aureus* [23] and dairy isolates of *Enterococcus faecalis* expressing different virulence factors [24].

Results showed that *bck1Δ* (MAPK kinase kinase (MAPKKK) mutant) and *slt2Δ* (MAPK mutant) were the most sensitive mutants to OG (viz., both MICs and MFCs = 25 μM; mean MICs and MFCs for other yeast strains = 47 and 50 μM, respectively; table data not shown). We further observed that CAS + OG chemosensitization could lower dosages of CAS and OG to achieve ≥99.9% fungal death, where *slt2Δ* required much smaller dosages of each reagent (CAS: 0.25 μg/mL; OG: 12.5 μM), when compared to the wild type (CAS: 2.00 μg/mL; OG: 25.0 μM) (Figure 2).

The *bck1Δ* and *slt2Δ* previously exhibited hypersensitivity to cell wall-perturbing agents including CAS [25] and, therefore, have been serving as screening tools for identifying new cell wall disrupting drugs [25]. Thus, hypersensitive response of *bck1Δ* and *slt2Δ* to OG indicates that OG could target the fungal cell wall system.

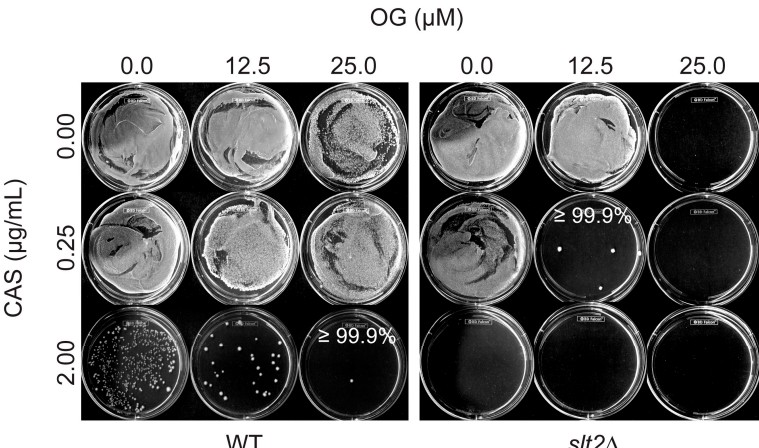

**Figure 2.** Exemplary chemosensitization (CAS + OG) test in *S. cerevisiae* wild type (WT) and *slt2Δ* strains. Results shown here are the determination of minimum fungicidal concentrations (MFCs), after MIC measurement in 96-well microplates, of antifungal agents (≥99.9% indicates achievement of ≥99.9% fungal death). Note that *slt2Δ* required much lower dosages of CAS and OG to achieve ≥99.9% fungal death, when compared to the WT (see the text).

### 3.2. Octyl Gallate Enhances the Efficacy of Caspofungin: Filamentous Fungi Bioassay

Antifungal chemosensitization (CAS + OG) was then investigated in filamentous fungal pathogens (*Aspergillus*, *Penicillium*). For FICIs in *Aspergillus*, "synergistic" FICI values (i.e., FICI ≤ 0.5) were found between OG and CAS for *A. flavus* 3357 and *A. parasiticus* 2999 (Table 2). Although there was no calculated synergism, as determined by "indifferent" [15] or "additive" interactions [16], there was enhanced antifungal activity of OG and CAS also in other *Aspergillus* strains, which was reflected in lowered MICs of OG or CAS (FICIs = 0.6 to 1.0) when two compounds were co-applied (Table 2). Of note, "synergistic" FICI values (FICI ≤ 0.5) were determined for most *Penicillium* strains tested (FICIs = 0.3 to 0.5). The only exception was *P. griseofulvum* 2159, where FICI was 0.6 (Table 2). Therefore, the results indicated that *Penicillium* species were more susceptible to OG-mediated chemosensitization than the *Aspergillus* strains examined.

**Table 2.** Antifungal chemosensitization of octyl gallate (OG; mM) to caspofungin (CAS; µg/mL) tested against filamentous fungi. Synergistic fractional inhibitory concentration indices (FICIs) and fractional fungicidal concentration indices (FFCI) (≤0.5) are shown in bold characters.[1.]

| Strains | Compounds | MIC Alone | MIC Combined | FICI | MFC Alone | MFC Combined | FFCI |
|---|---|---|---|---|---|---|---|
| *A. fumigatus* AF293 | CAS | 128 | 32 | 0.8 | 128 | 64 | 1.0 |
|  | OG | 0.2 | 0.1 |  | 0.4 | 0.2 |  |
| *A. fumigatus* sakAΔ | CAS | 128 | 8 | 0.6 | 128 | 64 | 1.0 |
|  | OG | 0.2 | 0.1 |  | 0.2 | 0.1 |  |
| *A. fumigatus* mpkCΔ | CAS | 128 | 8 | 0.6 | 128 | 64 | 0.8 |
|  | OG | 0.2 | 0.1 |  | 0.4 | 0.1 |  |
| *A. flavus* 4212 | CAS | 128 | 64 | 1.0 | 128 | 64 | 1.0 |
|  | OG | 0.2 | 0.1 |  | 0.4 | 0.2 |  |
| *A. flavus* 3357 | CAS | 128 | 2 | **0.5** | 128 | 128 | 2.0 |
|  | OG | 0.4 | 0.2 |  | 1.6 | 1.6 |  |
| *A. parasiticus* 5862 | CAS | 128 | 64 | 1.0 | 128 | 128 | 2.0 |
|  | OG | 0.4 | 0.2 |  | 1.6 | 1.6 |  |
| *A. parasiticus* 2999 | CAS | 128 | 4 | **0.5** | 128 | 64 | 0.6 |
|  | OG | 0.4 | 0.2 |  | 1.6 | 0.2 |  |

**Table 2.** *Cont.*

| Strains | Compounds | MIC Alone | MIC Combined | FICI | MFC Alone | MFC Combined | FFCI |
|---|---|---|---|---|---|---|---|
| Mean, *Aspergillus* [2] | CAS | 128.00 | 33.20 | 0.8 | 128.00 | 89.60 | 1.4 |
| | OG | 0.32 | 0.16 | | 1.12 | 0.76 | |
| *t*-test [3] | CAS | - | *p* < 0.001 | - | - | *p* < 0.05 | - |
| | OG | - | *p* < 0.05 | | - | *p* < 0.5 | |
| *P. expansum* W1 | CAS | 128 | 32 | **0.5** | 128 | 32 [4] | 0.8 |
| | OG | 0.2 | 0.05 | | 1.6 | 0.8 | |
| *P. expansum* FR2 | CAS | 128 | 32 | **0.5** | 128 | 32 | 0.8 |
| | OG | 0.2 | 0.05 | | 1.6 | 0.8 | |
| *P. expansum* W2 | CAS | 128 | 32 | **0.5** | 128 | 32 | 0.8 |
| | OG | 0.2 | 0.05 | | 1.6 | 0.8 | |
| *P. expansum* FR3 | CAS | 128 | 32 | **0.5** | 128 | 32 | 0.8 |
| | OG | 0.2 | 0.05 | | 1.6 | 0.8 | |
| *P. glabrum* 766 | CAS | 128 | 16 | **0.3** | 128 | 32 | **0.3** |
| | OG | 0.2 | 0.025 | | 1.6 | 0.05 | |
| *P. italicum* 983 | CAS | 64 | 16 | **0.5** | 64 | 16 | 0.8 |
| | OG | 0.2 | 0.05 | | 0.4 | 0.2 | |
| *P. griseofulvum* 2159 | CAS | 128 | 8 | 0.6 | 128 | 16 | **0.4** |
| | OG | 0.2 | 0.1 | | 0.8 | 0.2 | |
| *P. chrysogenum* 824 | CAS | 128 | 16 | **0.4** | 128 | 32 | **0.5** |
| | OG | 0.2 | 0.05 | | 0.2 | 0.05 | |
| Mean, *Penicillium* [2] | CAS | 117.33 | 20.00 | **0.4** | 117.33 | 26.67 | 0.6 |
| | OG | 0.20 | 0.05 | | 1.03 | 0.35 | |
| *t*-test [3] | CAS | - | *p* < 0.001 | - | - | *p* = 0.05 | - |
| | OG | - | *p* < 0.001 | | - | *p* < 0.05 | |
| Mean, TOTAL [2] | CAS | 122.18 | 26.00 | 0.6 | 122.18 | 55.27 | 1.0 |
| | OG | 0.25 | 0.10 | | 1.07 | 0.54 | |
| *t*-test [3] | CAS | - | *p* < 0.001 | - | - | *p* < 0.001 | - |
| | OG | - | *p* < 0.001 | | - | *p* < 0.05 | |

[1] OG was tested at 0.0125, 0.025, 0.05, 0.1, 0.2, 0.4, 0.8 mM, while CAS was examined at 0.0625, 0.125, 0.25, 0.5, 1, 2, 4, 8, 16, 32 µg/mL; [2] mean values were calculated by excluding mutant strains (*sakA*Δ, *mpkC*Δ, FR2, FR3); [3] *t*-test, Student's *t*-test for paired data (combined; chemosensitization) vs. mean MIC or MFC of each compound (alone; no chemosensitization) determined in strains; statistical analysis was performed according to "Statistics to use" [17], where *p* < 0.05 was considered significant; [4] achievement of 99.5% fungal death.

Regarding FFCIs in *Aspergillus*, enhanced fungicidality of CAS or OG was identified during chemosensitization (FFCIs = 0.6–1.0; additive [16]), despite no calculated synergism. For example, co-application of OG (0.1, 0.1 or 0.2 mM for *sakA*Δ, *mpkC*Δ or wild type, respectively) with CAS (64 µg/mL) achieved ≥99.9% fungal death of *A. fumigatus*, while individual treatment of each compound, alone, at the same concentrations allowed the survival of *A. fumigatus*. Of note, *sakA*Δ and *mpkC*Δ (antioxidant signaling mutants) [10,11] were more susceptible to the chemosensitizer, viz. they required a lower concentration of OG compared to the wild type (Table 2; Figure 3). However, the enhanced level of OG on CAS lethality was not high against these mutants when their MFC values were compared to that of the wild type, indicating that the chemosensitizing capability of OG was at the level of lowering MICs (but not MFCs, thus fungistatic, but not fungicidal) in *sakA*Δ and *mpkC*Δ. Meanwhile, no enhancement in fungicidality was identified in *A. flavus* 3357 and *A. parasiticus* 5862 during chemosensitization, even at the highest concentrations of either compound applied (FFCI = 2.0).

Synergistic FFCIs were also identified in *P. glabrum* 766, *P. griseofulvum* 2159 and *P. chrysogenum* 824 (FFCIs = 0.3 to 0.5) (Table 2; see also Figure S1). Although no calculated synergism was determined, there was enhanced antifungal activity of CAS and OG for the remaining *Penicillium* strains (FFCIs = 0.8; additive [16]) during chemosensitization. *P. glabrum* 766 and *P. chrysogenum* 824 were the most susceptible strains to the chemosensitization, where synergism was found for both

FICIs and FFCIs (Table 2; Figure S1). In general, *Penicillium* strains were more susceptible to CAS (viz., required lower concentration of CAS; 16–32 µg/mL) during chemosensitization, when compared to the *Aspergillus* strains (64–128 µg/mL CAS) (Table 2; Figure S1). The differences in susceptibility between *Penicillium* and *Aspergillus* to chemosensitization may be due to differences in cell wall sugar composition, such as mannose, galactose, galactofuranose, etc. [26,27]. Elucidation of the precise mechanism of differential susceptibility of fungi to chemosensitization warrants future in-depth investigation.

In the previous acute toxicity tests, the $LD_{50}$ values of OG in rat had been determined as 1960–2710 mg/kg body weight [28], while no observed adverse effect of OG was identified at 50 mg/kg body weight/day in a reproductive toxicity study [29]. Therefore, the mean MFCs for OG, viz. 0.76 mM (0.215 mg/mL) or 0.35 mM (0.099 mg/mL), applied to achieve ≥99.9% fungal death in *Aspergillus* or *Penicillium*, respectively, are considered safe (See Table 2).

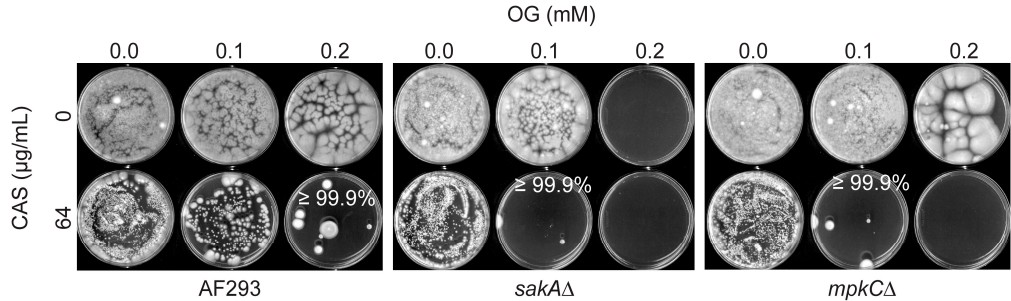

**Figure 3.** Chemosensitization test in *A. fumigatus* wild type (AF293), *sakA*Δ and *mpkC*Δ. Results shown here are the determination of MFCs of antifungal agents (CAS, OG) (≥99.9% indicates achievement of ≥99.9% of fungal death).

### 3.3. Octyl Gallate Debilitates Antioxidant Mutants during Chemosensitization

The mode of antifungal action of OG has been discussed in prior studies, where: (i) OG interrupts or disorganizes the lipid bilayer-protein interface in fungal cells [30]; and (ii) OG functions as a pro-oxidant (redox-active oxidative stressor), thus triggering cytotoxicity in fungi [31]. We speculate that, in addition to destabilizing cell wall integrity, disruption of cellular components by the pro-oxidant characteristic of OG could also be one mechanism of action for the enhancement of CAS activity during OG-mediated chemosensitization.

For example, the *A. fumigatus* *sakA*Δ and *mpkC*Δ antioxidant mutants were more susceptible to chemosensitization (OG + CAS) compared to the wild type (see above). Redox-active compounds, such as benzo derivatives or sulfur-containing compounds, could function as potent redox-cyclers in microbes and, thus, inhibit pathogen growth by interfering with cellular antioxidant systems, redox homeostasis or the function of redox-sensitive macromolecules [32,33]. Therefore, it is postulated that, in addition to destabilizing the cell wall integrity system, the redox-active OG (chemosensitizer) can further debilitate the susceptibility, viz. defects in ameliorating oxidative stress and/or disruption of cellular redox homeostasis, of the antioxidant mutants during chemosensitization. From the pathogens' perspective, an intact antioxidant signaling system, such as the MAPK pathway, plays an important role in fungal defense against the OG-mediated chemosensitization.

Notably, previous studies showed that, in addition to the cell wall integrity system, another signaling pathway, viz. the "antioxidant" MAPK system, also plays an important role in fungal susceptibility to cell wall-interfering agents (see below). In principle, a functionally-intact antioxidant MAPK system is required for achieving the fungicidal effects of cell wall-disrupting drugs, while mutations in the system result in resistance to the drugs. For instance, the antioxidant MAPK pathway mutants of *S. cerevisiae*, such as *hog1* (MAPK) or *pbs2* (MAPK kinase; MAPKK) mutants, exhibited tolerance to cell wall-interfering agents [34–36]. A similar type of drug tolerance was also

observed in *Candida albicans* [37]. Fungal dialogs between "antioxidant" and "cell wall integrity" MAPK pathways have been well documented recently [38]. Identification of the precise mechanism or cellular target(s) of OG during chemosensitization warrants future study.

## 4. Conclusions

In conclusion, chemosensitization could be an effective antifungal intervention strategy (see also [39]). OG, a safe, alkyl derivative of natural benzoate, possesses the potential to serve as an antimycotic chemosensitizer when co-applied with CAS. This potential appears to be greatest with *Penicillium* strains. OG-mediated chemosensitization, as presented here, can sensitize cell wall integrity and antioxidant systems of filamentous fungi and, thus, can lower effective doses of toxic antifungal agents (such as CAS), leading to coincidental lowering of environmental and health risks. The use of safe chemosensitizers as intervention catalysts that debilitate filamentous fungal pathogens could be a viable approach for pathogen control. Inclusion of more clinical strains, such as *A. fumigatus*, *A. terreus*, *A. niger*, etc., in future tests would be necessary. There could be significant differences in efficacy for different strains of the same species. Future in vivo studies are also necessary to determine potential applications of chemosensitization in therapeutic settings.

**Supplementary Materials:** The following are available online at http://www.mdpi.com/2571-8800/1/1/4/s1, Figure S1: Chemosensitization test in *A. flavus* 4212, *A. parasiticus* 2999, *P. italicum* 983 and *P. glabrum* 766; Table S1: The model yeast *Saccharomyces cerevisiae* strains used in this study.

**Author Contributions:** J.H.K. conceived of, designed and performed the research including data analysis and interpretation and the literature search and wrote the manuscript. K.L.C. performed the antifungal bioassays and prepared the figures. L.W.C., Research Leader, directed the research.

**Acknowledgments:** This research was conducted under USDA-ARS CRIS Project 5325-42000-039-00D.

**Conflicts of Interest:** The authors declare no conflict of interest.

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
