# Peer review of "Octyl Gallate as an Intervention Catalyst to Augment Antifungal Efficacy of Caspofungin"

_2571-8800, doi:10.3390/j1010004_

Round 1

Reviewer 1 Report

Dear Authors,

In my opinion, manuscript ID J-305215 entitled: “Octylgallateas an intervention catalyst to augment antifungal efficacy of caspofungin” by Jong H. Kim, Kathleen L. Chan and Luisa W. Cheng is within the scope of Journal of Multidisciplinary Scientific Journal.

Fungal infections have increased in frequency in the last decades and can range in severity from superficial to life-threatening. Nowadays, there are some classes of antifungal drugs, among others echinocandins, which are attractive treatment options for the treatment of select invasive fungal infections, most notably invasive candidiasis and treatment-refractory invasive aspergillosis. Caspofungin is echinocandin with clinical use due to their good solubility, antifungal spectrum, and pharmacokinetic properties. Additionally, antifungal chemosensitization of caspofungin (as presented here) could be an effective antifungal intervention strategy. Conclusions are very interesting.

The article is very well organized and written. The results are presented in the form of schemes, tables and figures. Therefore, I accept this manuscript in present form.

Best regards,

Author Response

Dear Editor,

We appreciate the reviewers’ valuable and constructive comments for our submitted manuscript. We carefully revised our manuscript according to the reviewers’ comments and recommendation, and are re-submitting the revised version of the manuscript to the MDPI journal J. Please let us know should you have any questions or require further information.

Thank you very much.

Sincerely,

Jong H. Kim

RESPONSE TO REVIEWER’S COMMENTS:

Reviewer: 1

Authors’ Response: We appreciate the reviewer’s valuable comments. We incorporated the following sentence provided by the reviewer into the INTRODUCTION section (Page 1, lines 35 - 36, highlighted) of the revised manuscript:

“CAS is in clinical use due to its good solubility, antifungal spectrum, and pharmacokinetic properties.”

Reviewer 2 Report

In this manuscript, the authors report the effects of adding Octyl Gallate as a chemosensitizing agent to increase the antifungal efficacy of Caspofungin.  Further, using fungal antioxidant mutants, they showed increased sensitivity to inhibition.  This communication is well written and certainly worthy of publication.  This reviewer suggests acceptance after the following minor changes:

It is stated that Octyl Gallate is considered safe for consumption and is used as a food additive.   However, are the antifungal concentrations reported in this manuscript  anywhere near the concentrations known to be safe?  Some indication of what is considered safe and how it compares to the concentrations used here is warranted.  (Is anything like an LD50 known?)

Another minor weakness is the minimal number of fungi tested for some of the strains reported.  In particular, there is only data for one wild type A. fumigatus strain and most of the Penecillin strains.  While the number of different fungi do help solidify the point, there can be significant differences in efficacy for different strains of the same species.  It would significantly strengthen the paper to have at least a few more strains, in particular for the A. fumigatus to really show how different the mutants are.  

Author Response

Dear Editor,

We appreciate the reviewers’ valuable and constructive comments for our submitted manuscript. We carefully revised our manuscript according to the reviewers’ comments and recommendation, and are re-submitting the revised version of the manuscript to the MDPI journal J. Please let us know should you have any questions or require further information.

Thank you very much.

Sincerely,

Jong H. Kim

RESPONSE TO REVIEWER’S COMMENTS:

Reviewer: 2

Comment 1: It is stated that Octyl Gallate is considered safe for consumption and is used as a food additive. However, are the antifungal concentrations reported in this manuscript anywhere near the concentrations known to be safe?  Some indication of what is considered safe and how it compares to the concentrations used here is warranted.  (Is anything like an LD50 known?)

Authors’ Response 1: We appreciate the reviewer’s valuable comments. We incorporated the following information including new references in the text (Page 7, lines 195 - 199, highlighted):

“In the previous acute toxicity tests, the LD50 values of OG in rat have been determined as 1,960 - 2,710 mg/kg body weight [28], while no observed adverse effect of OG was identified at 50 mg/kg body weight/day in a reproductive toxicity study [29]. Therefore, the mean MFCs for OG, viz., 0.76 mM (0.215 mg/mL) or 0.35 mM (0.099 mg/mL), applied to achieve ≥ 99.9% fungal death in Aspergillus or Penicillium, respectively, are considered safe (See Table 2).”

Comment 2: Another minor weakness is the minimal number of fungi tested for some of the strains reported.  In particular, there is only data for one wild type A. fumigatus strain and most of the Penecillin strains.  While the number of different fungi do help solidify the point, there can be significant differences in efficacy for different strains of the same species. It would significantly strengthen the paper to have at least a few more strains, in particular for the A. fumigatus to really show how different the mutants are.  

Authors’ Response 2: We agree with the reviewer’s valuable suggestion. In fact, we are planning to perform comprehensive chemosensitization study by including more clinical strains, which will be obtained from university hospitals and US Center for Disease Control and Prevention. We will present the data from the new investigation in the next publication. According to the reviewer’s advice, we incorporated the following sentences in the text (Page 8, lines 237 - 240, highlighted):

“Inclusion of more clinical strains, such as A. fumigatus, A. terreus, A. niger, etc., in the future tests would be necessary. There can be significant differences in efficacy for different strains of the same species. Future in vivo studies are also necessary to determine potential application of chemosensitization for therapeutic settings.”
